# The Early Impact of the Covid-19 Emergency on Mental Health Workers: A Survey in Lombardy, Italy

**DOI:** 10.3390/ijerph17228615

**Published:** 2020-11-20

**Authors:** Filippo Rapisarda, Martine Vallarino, Elena Cavallini, Angelo Barbato, Camille Brousseau-Paradis, Luigi De Benedictis, Alain Lesage

**Affiliations:** 1Research and Development Team, Sociosfera ONLUS SCS, Via Antonio Gramsci 8, 20831 Seregno, Italy; 2Department of Brain and Behavioral Sciences, University of Pavia, 27100 Pavia, Italy; martine.vallarino@unipv.it (M.V.); elena.cavallini@unipv.it (E.C.); 3IRCCS Istituto di Ricerche Farmacologiche Mario Negri, Via Mario Negri 2, 20156 Milano, Italy; angelo.barbato@marionegri.it; 4Institut Universitaire en Santé Mentale de Montréal, CIUSSS de l’Est-de-l’Île-de-Montréal, 7401 Rue Hochelaga, Montreal, QC H1N 3M5, Canada; camille.brousseau-paradis@umontreal.ca (C.B.-P.); ldebenedictis.crfs@ssss.gouv.qc.ca (L.D.B.); alesage.iusmm@ssss.gouv.qc.ca (A.L.)

**Keywords:** Covid-19, mental health service, workers, burnout

## Abstract

Lombardy was the epicenter of the Covid-19 outbreak in Italy, and in March 2020 the rapid escalation in cases prompted the Italian Government to decree a mandatory lockdown and to introduce safety practices in mental health services. The general objective of the study is to evaluate the early impact of the Covid-19 emergency and quarantine on the well-being and work practices of mental health service personnel and professionals. Data were collected through an online survey of workers and professionals working with people with mental health problems in Lombardy in several outpatient and inpatient services. Their socio-demographic characteristics, professional background, description of working conditions during lockdown and psychological distress levels were collected. All analyses were performed on a sample of 241. Approximately, 31% of the participants obtained a severe score in at least one of the burnout dimensions, 11.6% showed moderate or severe levels of anxiety, and 6.6% had a moderate or severe level of depression. Different work conditions and patterns of distress were found for outpatient service workers and inpatient service workers. The overall impact of the Covid-19 emergency on mental health workers’ level of distress was mild, although a significant number of workers experienced severe levels of depersonalization and anxiety. More research is needed to assess specific predictive factors.

## 1. Introduction

Italy was one of the first countries to be severely affected by the global Covid-19 pandemic. The epicenter of the outbreak was located in the Lombardy region, the largest and most densely populated region of Italy with a total population of 10.06 million inhabitants in 2020, which is one-sixth of the Italian population. The first Covid-19 cases in Lombardy were officially reported in the middle of February 2020, and the rapid escalation of the outbreak required the Italian Government to introduce a mandatory lockdown on 7 March 2020. By the end of September 2020, the National Institute of Health reported 106,421 Covid-19 cases and 16,937 Covid-19related deaths in Lombardy (www.epicentro.iss.it), which represented approximately 34% of Italian Covid-19 cases and 47% of Italian Covid-19-related deaths.

Healthcare workers had to face a new disease as well as its potential effects on their mental health, including burnout and symptoms of depression, anxiety, insomnia, grief and the development of post-traumatic stress disorder [1,2]. Deterioration in workers’ mental health may result from an increase in work-related stress caused by the need to change work procedures and to adapt to a changed work environment, as well as exposure to traumatic experiences [3,4].

Some reports on the effects of the pandemic outbreak on people with mental disorders and on mental health services (MHSs) have been published [5,6,7,8]. The risk of Covid-19 infection and the consequent restrictive measures introduced in Italy by the health authorities in March 2020 deeply affected the work practices of MHSs. Activities in MHSs were reduced in terms of opening hours and the number of contacts and interventions provided [9]. Some inpatient units were organized with dedicated areas for Covid-19 positive patients with acute mental disorders [10], with home care and off-site activities being provided for urgent cases, as well as the implementation of remote psychosocial interventions [8].

We can hypothesize that the overall state of emergency and changes in service practices could have an impact on mental health workers (MHWs) wellbeing; however, no investigation on this issue has been carried out to date. In fact, despite the amount of research that addresses various aspects of the impact of Covid-19 on health care workers, to our knowledge no study specifically targeting mental health professionals has been published. Therefore, to fill this gap, we designed this study to investigate the early impact of the emergency and quarantine on the well-being, work conditions and work practices of MHS personnel and professionals in Lombardy and to compare the findings with available data on healthcare workers facing the Covid-19 outbreak.

## 2. Materials and Methods

### 2.1. Design and Procedure

The present study was configured as a population survey of workers and professionals working with people with mental health problems in the Lombardy Region in Italy. Participants were recruited from different outpatient and inpatient services including community mental health centers, residential facilities, hospital wards, and nursing homes. Consultants and freelance professionals, and staff operating in private or nonprofit centers and residential facilities were also included. Participants who were not actively working during the data collection phase (i.e., on temporary leave because of the lockdown) were excluded from the data analysis.

Data were collected through an online platform and recruitment was carried out by spreading a link to an invitation to participate through various professional networks. A “snowball” dissemination strategy was promoted, in which each participant was asked to send the survey invitation link to their colleagues. Two strategies were adopted to increase the validity of the sampling method: during the first week of data collection, investigators sent the survey invitation to directors and service coordinators of public, nonprofit or private services with a request for them to spread it to the multi-professional teams that they supervised. Then, after the first week of data collection, the current preliminary sample was compared with the final sample from a similar survey of mental health workers conducted in 2015 [11], and the investigators tried to correct and compensate for over- and under-represented professional categories by sending targeted invitations to members of specific professional groups (i.e., nurses).

Upon acceptance of informed consent, the subjects were asked to fill in an anonymous online questionnaire that lasted about 10 minutes. Data collection took place from 15 April to 15 May 2020.

### 2.2. Instruments

The survey was made up of four sections. Section 1 included questions, partially retrieved from a previous survey [11], aimed at gathering socio-demographic and professional background information. Section 2 and Section 3 were composed of ad hoc questions designed to collect information about work conditions during the emergency (i.e., changes in workload, experience of Covid-19 infection among users and colleagues, perceived probability of infection at work and concerns) and the provision of interventions. Ad hoc items investigated different kinds of interventions, including whether they were provided during the outbreak and how they were provided (direct or remote). Section 4 included three validated questionnaires on psychological distress (the Maslach Burnout Inventory, Generalized Anxiety Disorder-7 and Patient Health Questionnaire-9).

The Maslach Burnout Inventory (MBI) [12,13,14] is a self-administered instrument to assess burnout in health organizations through three subscales, i.e., Emotional Exhaustion, Depersonalization, and (reduced) Personal Accomplishment. Items are framed as statements of job-related feelings, which are rated on a 7-point frequency scale ranging from 0 (never) to 6 (every day). In the present study, only the Exhaustion and Depersonalization scales were adopted, since the developers of the MBI have expressed some doubts about whether Personal Accomplishment is a valid burnout dimension [15,16]. Moreover, following the example of Bianchi et al. [17], the Emotional Exhaustion and Depersonalization subscales were combined to obtain a global burnout index.

The Generalized Anxiety Disorder-7 questionnaire (GAD-7) [15] contains seven items and it has been used to assess anxiety levels in the general population [18,19] and also in healthcare workers during the Covid-19 pandemic [3,20].

The Patient Health Questionnaire-9 [21,22] is a validated questionnaire that assesses the presence of depressive states in general, clinical or professional populations. We adopted the Italian version validated by Mazzotti et al. [23], which has been used for healthcare staff surveys during the outbreak in Italy [20].

### 2.3. Data Analysis

Descriptive statistics were computed for sociodemographic and work-related variables, and the differences between outpatient and inpatient services were assessed through parametric and nonparametric tests according to the nature and distribution of the variables analyzed.

Some response categories were merged during data analysis:Setting: in the “outpatient service” category we included community mental health centers, office-based activities, counselling and psychotherapy services, day hospitals and day-care units; in the “inpatient service” category we included hospital wards, residential facilities (including those with high intensity services to those with low intensity services without staff onsite) and nursing homes.The category of “counsellor” was adopted for rehabilitation workers with a university degree, i.e., professional educators, occupational therapists and rehabilitation technicians.

Moreover, we estimated the number of participants that scored severe levels in each MBI subscale according to the cutoff levels proposed by the Italian MBI manual [14], the number with clinical levels of anxiety (GAD-7 ≥ 10) according to Spitzer et al. [18] and those with depression (PHQ-9 ≥ 10) according to Gilbody et al. [24].

Generalized linear models were used to explore predictors of burnout (Emotional Exhaustion + Depersonalization), anxiety (GAD-7) and depressive (PHQ-9) states. Several models were tested using the socio-demographics and work-related variables.

All data analysis was performed using IBM SPSS Statistics v. 23.

### 2.4. Ethical Issues

This study was approved by the Ethical Board Committee of the University of Pavia with protocol number 49/2020, and meets all ethical and legal standards applicable to research using this survey modality. The acquisition of data was carried out through the online platform LimeSurvey (https://www.limesurvey.org/), which operates in compliance with the European Regulation (EU) 2016/679-General Data Protection Regulation, art 26. The inclusion of subjects was conditional on them having read the information on the processing of personal data for research and the acceptance of informed consent.

## 3. Results

### 3.1. Sample Description

Of the 337 participant who began the online survey, 40 (11.9%) then declared that they had suspended all professional activities during the Covid-19 emergency, and so they were excluded from the present study. The rate of the suspension of job activities was significantly higher for peer supporters (51.5%), and ranged between 0% and 15.4% for other professional roles. A further 56 participants dropped out of the online survey after completing the socio-demographic section but before they had answered any job-related questions, thus they were excluded from the analysis. Therefore, the analyses were performed on a sample of 241 participants.

The sociodemographic and professional characteristics of the sample are shown in Table 1. Approximately two-thirds of the participants worked in outpatient services, most of them were female (76.8%), with an average age of 44.2 years (sd = 12.3) and a university degree (87.9%). Psychologists and counsellors were the most represented professional category, medical doctors (mostly psychiatrists), nurses and social workers were also adequately represented. Most of participants worked only with adult users/clients in public services or in community and residential facilities run by non-profit organizations. Significant differences between outpatient and inpatient services are reported in the table.

### 3.2. Work Conditions during the COVID-19 Outbreak

Table 2 presents descriptive statistics with regard to the participants’ work conditions during the COVID-19 outbreak. While more than two-thirds of participants working in outpatient services reported a decrease in the workload (68.2%), for most of their colleagues working in inpatient services, the workload remained the same (37.9%) or increased (41.4%). In outpatient services, more than half of the workers maintained only remote contact with users/clients (56.5%), whereas in inpatient services almost all workers (94.6%) continued to have direct contact with users. More than half of the participants reported having colleagues (54.8%) or users/clients (50.2%) infected by COVID-19, even though a significant percentage (19.9–21.6%) indicated that no official diagnosis had been made. The perceived risk of contracting the virus at work was higher for inpatient service workers, who also declared higher levels of concern of being infected and of transmitting infection to users/clients.

Even though most of the participants stated that they were equipped with adequate personal protective equipment or that they did not need it, a significant percentage of workers (21.5%) declared that they were inadequately equipped. A small percentage of the participants (7.9%) received psychological support provided by their organization, and it was available to less than half of the participants (41.9%).

### 3.3. Psychological Distress

#### 3.3.1. Burnout

Participants’ MBI-subscale scores are reported in Table 3. No statistically significant differences were detected between outpatient and inpatient services for burnout dimensions. Twenty percent of participants obtained a severe score in at least one of the burnout dimensions.

#### 3.3.2. Anxiety and Depression

The mean anxiety score for the overall sample was 5.1 (sd = 3.4), and 11.6% of participants scored moderate or severe anxiety levels and there were statistically significant differences between outpatient and inpatient services’ staff. Participants indicated that “feeling nervous” and “trouble relaxing” were the most common anxiety-related problems. Slightly significant statistical differences were also found between outpatient and inpatient services workers for item 1, item 2 and item 5. Participants’ PHQ-9 score for the overall sample was 4.7 (sd = 2.9), with only a small percentage (6.6%) scoring moderate or severe levels of depression. “Sleeping problems” and “feeling tired, lack of energy” were the most common depression-related problems.

#### 3.3.3. Correlation between Scores of Psychological Distress

A moderately significant correlation was found between GAD-7 and PHQ-9 total scores (r = 0.68; *p* < 0.01). Moreover, the GAD-7 total score was also significantly correlated with the MBI Exhaustion scale (r = 0.53; *p* < 0.01) and the MBI Depersonalization scale (r = 25; *p* < 0.01), and the same correlation pattern was found for PHQ-9 total score with MBI Exhaustion (r = 0.54; *p* < 0.01) and MBI Depersonalization (r = 38; *p* < 0.01). A moderate correlation (r = 0.41; *p* < 0.01) was also found between Exhaustion and Depersonalization.

#### 3.3.4. Predictors of Psychological Distress

Table 4 shows the results of three general linear models predicting burnout, GAD-7 and PHQ-9 total scores as dependent variables. In the first linear model, being a woman, working in close contact with Covid-19-infected users, being a medical doctor, working in outpatient services and perceiving a medium or high risk of contracting Covid-19 at work were conditions associated with burnout. A second model identified being a woman and perceiving a medium or high probability of contracting COVID-19 at work as positively associated with higher GAD-7 scores. For PHQ-9, being a woman and perceiving a high probability of contracting Covid-19 at work were associated with higher depression scores, but working with non-infected users had a slight negative association. Age (as a covariate), the availability of PPE, perceived change in workload and the availability of staff psychological support were included in the models, but showed no statistically significant effect.

### 3.4. The Provision of In-Person Versus Remote Interventions

Table 5 shows the different interventions that were performed during the early stage of the Covid-19 outbreak in outpatient and inpatient services. While most interventions were still delivered despite the emergency, home visits, visits in the community and group activities (in outpatient settings) were reduced. In outpatient services, individual counselling, psychotherapy and individual rehabilitation, and meetings with users’ family members were mainly provided remotely. Conversely, in inpatient settings these interventions were provided through direct contact with users, with the exception of meetings with family members, which were done remotely. In outpatient services, back office tasks and staff supervision were performed remotely, but in inpatient services, the same management duties were performed onsite. Only a small percentage of participants had been assigned to personally assist with hygiene and meals and drug administration, interventions that were delivered mostly in person.

## 4. Discussion

This study examined some relevant aspects of the work experience and psychological distress of MHS personnel and professionals in Lombardy during the COVID-19 outbreak. In our sample, significant differences were found between outpatient and inpatient services’ staff. In outpatient services, staff reported an overall decrease in the workload with a significant amount of online or telephone contact with users and clients, whereas in inpatient settings, the workload increased and most of the activities continued to rely on direct contact with users, which implied more concerns regarding infection and a rise in psychological distress, tension and restlessness. Our finding that the perceived workload increased during the outbreak in inpatient setting diverges from that of Clerici et al. [25], who recently reported that they found a reduction in psychiatric ward admission rates in Lombardy during March 2020. However, we evaluated perceived workload from a MHW perspective, and we hypothesize that changes in procedures, reductions in onsite personnel and higher levels of patients’ severity of illness during the emergency lockdown may have created additional workload and stress despite the decrease in caseload.

In our sample of MHWs, severe levels of depersonalization were more frequent than emotional exhaustion and involved one in five workers. This rate was slightly higher than findings from a Spanish survey of healthcare workers during the earliest stage of the Covid-19 outbreak in April 2020 [26]. We can hypothesize that protective procedures, fear of infection and remote working may have a role in fostering feelings of detachment from clients. Moreover, feelings of detachment only partially overlapped with severe anxious or depressive reactions, and may be a defensive psychological mechanism to cope with the fear of infection and overall stress. However, more studies will be necessary to compare our findings with burnout scores in the Italian context and to validate this hypothesis.

Mild levels of depressive and anxiety symptoms were also found. The most frequent symptoms reported by our participants were a combination of tension and tiredness that could indicate that during the pandemic, MHWs’ distress was related to stressful working conditions. The mean total PHQ-9 score that was found in our study is similar to the results of Buselli et al. [20], who found a mean PHQ-9 score of 4.5 (sd = 6.4) in healthcare workers. Anxiety was higher for inpatient services’ staff, and this result could be linked to increased concern of infection, as presented in Table 2. Fear of infection was reported by Lai [3] as a source of professional stress in medical personnel in China.

Our findings also partially replicate some of the risk factors found in previous studies: distress levels were higher in women as found in previous studies [3,20]; direct exposure to Covid-19-infected users increased psychological distress with regard to burnout, but not anxiety or depression; and medical doctors seem to be more vulnerable to burnout. In our findings, a crucial role was played by the perceived probability of infection, a cognitive factor that was associated with burnout, anxiety and depression, in other words, distress was not necessarily associated with real risk (a result of direct contact with infected users), but was related to the cognitive appraisal of risk. Conversely, the presence of higher emotional distress may enhance the risk perception [27]. Moreover, we also found that, for MHWs, having direct contact with non-infected clients had a mild, protective effect on depressive states compared to working remotely. Maintaining face-to-face contact with users may increase anxiety or feelings of depersonalization (as a coping strategy), but slightly helped vulnerable workers from feeling depressed during lockdown.

Our results showed how MHWs adopted remote work practices. Tele-psychiatry practices have been tested in different contexts and can be useful and effective for both psychiatric assessment and monitoring interventions [28,29], individual psychotherapy [30,31] and group interventions [32]. Online psychological support programs have been reported in several countries during the Covid-19 pandemic, including China [33], Germany [34], the United States [35], and France [36]. However, our data suggest that remote interventions are applied less in inpatient settings, where the onsite presence of staff is required despite the pandemic emergency, and users require more intense assistance with daily activities, drug administration and social rehabilitation [37].

The findings of this study should be interpreted in light of several limitations. The first limitation is that the composition of the sample composition only partially replicates the typical distribution of professional roles among Italian MHSs. This is due to the snowball convenience sampling strategy. The study sample has a larger percentage of psychologists and peer support workers compared to a previous survey that collected data directly from mental health teams from public and nonprofit services [11]. The number of peer support workers in the current sample is over-representative, and this is probably a consequence of a sampling bias caused by the involvement of a peer support association (a factor that we could evaluate positively as an index of service users’ involvement in research), while the relatively large number of psychologists compared to 2015 is a consequence of the investigators’ choice to include freelance professionals that work in private practice in the 2020 survey. A further limitation was the decision to exclude MHWs and professionals who stopped work during the study period from the survey. Although this decision was taken in order to focus on working conditions, it did not allow us to test the impact of the emergency situation on the psychological wellbeing of MHWs who stopped working for any reason. A third methodological limitation is related to the cross-sectional design, which does not allow us to evaluate the change in psychological distress within subjects before and after the emergency period.

## 5. Conclusions

This is the first survey on the early impact of the pandemic on MHWs and the results will be compared with other studies emerging from other countries to confirm the findings. In general, the early impact of the Covid-19 emergency on mental health workers in terms of anxiety and depression was mild, but one in three workers experienced severe levels of burnout. However, more research is needed to assess the specific predictive factors. Both general and job-specific stressors are at play, but workplace protectors such as increases in professional roles and social support may have mitigated the impact. The longer term consequences on staff should be assessed together with a more accurate conceptualization of the psychological effect of fear of infection, remote working and safety procedures.

## Figures and Tables

**Table 1 ijerph-17-08615-t001:** Sample description.

	Outpatient Service N = 154	Inpatient Service N = 87		Total Sample N = 241
**Sex**				
Female	123 (79.9%)	62 (71.3%)		185 (76.8%)
**Age**			**	
Mean. (sd)	46.5 (12.0)	40.1 (11.7)		44.2 (12.3)
**Education**			**	
Professional school or lower	7 (4.2%)	13 (14.9%)		20 (6.3%)
High school	5 (3.2%)	9 (10.3%)		14 (5.8%)
Bachelor of arts	27 (17.5%)	26 (29.9%)		53 (22.0%)
Master’s degree	33 (21.4%)	10 (11.5)		43 (17.8%)
Medical specialization/PhD/Other	82 (53.2%)	29 (33.3%)		111 (46.1%)
**Professional role**			**	
Psychologist	65 (42.2%)	8 (9.2%)		73 (30.3%)
Counsellor	33 (21.4%)	35 (40.2%)		68 (28.2%)
Medical doctor	15 (9.7%)	13 (14.9)		28 (11.6%)
Social worker	15 (9.7%)	0 (0.0%)		15 (6.2%)
Nurse	11 (7.1%)	16 (18.4%)		27 (11.2%)
Peer supporter	6 (3.9%)	3 (3.4%)		9 (3.7%)
Support worker	1 (0.6%)	6 (6.9%)		7 (2.9%)
Manager/coordinator	3 (1.9%)	4 (4.6%)		7 (2.9%)
Other	5 (3.2%)	2 (2.3%)		7 (2.9%)
**Users age**			**	
Adult users	110 (71.4%)	81 (93.1%)		191 (79.3%)
Adult and underage users	24 (15.6%)	2 (2.3%)		26 (10.8%)
Children and adolescent users	20 (13.0%)	4 (4.6%)		24 (10.0%)
**Labor contract**			*	
Employee	75 (48.7%)	73 (83.9%)		148 (61.4%)
Consultant	73 (47.4%)	12 (13.8%)		85 (35.3%)
Apprentice	1 (0.6%)	2 (2.3%)		3 (1.2%)
Volunteer	5 (3.2%)	0 (0.0%)		5 (2.1%)
**Who runs the service/structure?**			**	
Public service	75 (48.7%)	33 (37.9%)		108 (44.8%)
Non-profit organization	29 (18.8%)	28 (32.2%)		57 (23.7%)
Private organization	37 (24.0%)	7 (8.0%)		14 (5.8%)
Private service contracted with public service	13 (8.4%)	19 (21.8%)		32 (13.3%)

* = *p* < 0.05; ** = *p* < 0.01.

**Table 2 ijerph-17-08615-t002:** Mental health workers’ work conditions during the COVID-19 outbreak.

	Outpatient Service N = 154	Inpatient Service N = 87		Total Sample N = 241
**Workload variation due to the emergency**				
No changes	33 (21.4%)	33 (37.9%)	**	66 (27.4%)
The workload decreased	105 (68.2%)	16 (18.4%)		121 (50.2%)
The workload increased	15 (9.7%)	36 (41.4%)		51 (21.2%)
* Missing*	*1 (0.6%)*	*2 (2.3%)*		*3 (1.2%)*
**Do any of your colleagues got COVID-19?**			**	
None	85 (55.2%)	24 (27.6%)		109 (45.2%)
Probably. but have not received an official diagnosis	30 (19.5%)	22 (25.3%)		52 (21.6%)
Yes, they have been cured at home	28 (18.2%)	31 (35.6%)		59 (24.5%)
Yes, they have received inpatient care	11 (7.1%)	10 (11.5%)		21 (8.7%)
**Do any of your users/clients got COVID-19?**			-	
None	77 (50.0%)	43 (49.4%)		120 (49.8%)
Probably, but have not received an official diagnosis	37 (24.0%)	11 (12.6%)		48 (19.9%)
Yes, they have been cured at home or in our service	15 (9.7%)	14 (16.1%)		29 (12.0%)
Yes, they have received inpatient care	25 (16.2%)	19 (21.8%)		44 (18.3%)
**Levels of contact with users/clients**			**	
Didn’t have direct contacts with users/clients	88 (57.1%)	5 (5.7%)		93 (38.6%)
Had direct contacts with non-infected users/clients	24 (15.6%)	40 (46.0%)		64 (36.5%)
Had direct contacts with covid-19-infected users-clients	42 (27.3%)	42 (42.3%)		84 (34.8%)
**Perceived risk of contracting covid19 at work**			**	
Low probability	92 (59.7%)	19 (21.8%)		111 (46.1%)
Medium probability	53 (34.4%)	52 (59.8%)		105 (43.6%)
High probability	9 (5.8%)	16 (18.4%)		25 (10.4%)
**How much are you worried of being** **infected by COVID-19 at work?**			**	
Not at all	51 (33.1%)	6 (6.9%)		57 (23.7%)
A little worried	60 (39.0%)	33 (37.9%)		93 (38.6%)
Somewhat worried	39 (25.3%)	45 (51.7%)		84 (34.9%)
Very worried	4 (2.6%)	3 (3.4%)		7 (2.9%)
**How much are you worried of infecting users with COVID-19?**			**	
Not at all	69 (44.8%)	8 (9.2%)		77 (32.0%)
A little worried	52 (33.8%)	29 (33.3%)		81 (33.6%)
Somewhat worried	27 (17.5%)	35 (40.2%)		62 (25.7%)
Very worried	6 (3.9%)	15 (17.2%)		21 (8.7%)
**Are you always equipped with** **personal protective equipment (PPE)?**			**	
Yes, always	85 (55.2%)	71 (81.6%)		156 (64.7%)
Never or not always	39 (25.3%)	13 (14.9%)		52 (21.5%)
PPE not required for my current job	30 (19.5%)	2 (2.3%)		32 (13.3%)
**Did you receive psychological support in your service/organization?**			**	
Not available in my service	106 (68.8%)	34 (39.1%)		140 (58.1%)
Available but I did not request it	38 (24.7%)	44 (50.6%)		82 (34.0%)
Yes. I received psychological support at work	10 (6.5%)	9 (10.3%)		19 (7.9%)

* *p* = < 0.05; ** = *p* < 0.01.

**Table 3 ijerph-17-08615-t003:** Psychological distress of mental health workers.

	Outpatient Service	Inpatient Service		Total Sample
***MBI–Burnout***				
Mean. Sd	16.8 (12.1)	16.6 (10.5)		16.7 (11.5)
Above the severe cut off ^a^ level. n (%)	45 (31.9%)	26 (31.3%)		71 (31.7%)
***MBI—Exhaustion scale***				
Mean. Sd	13.8 (9.9)	13.7 (8.9)		13.7 (9.5)
Above the severe cut off level. n (%)	22 (14.3%)	11 (12.6%)		33 (13.7%)
***MBI–Depersonalization scale***				
Mean. Sd	3.0 (3.8)	2.9 (3.4)		3.0 (3.6)
Above the severe cut off level. n (%)	31 (20.1%)	17 (19.5%)		48 (19.9%)
***GAD-7***				
Total GAD-7 score, mean (sd)	4.5 (3.0)	6.0 (3.8)	**	5.1 (3.4)
Staff above the “moderate anxiety” cut off ^b^, n (%)	14 (9.1%)	14 (16.1%)	**	28 (11.6%)
***PHQ-9***				
Total PHQ-9 score, mean (sd)	4.5 (2.8)	5.1 (3.1)		4.7 (2.9)
Staff above the “moderate depression” cut off ^c^, n (%)	9 (5.8%)	7 (8.0%)		16 (6.6%)

* *p* = < 0.05; ** = *p* < 0.01.; ^a^ = above severe level in Exhaustion or Depersonalization scales; ^b^ = GAD-7 score ≥ 10; ^c^ = PHQ-9 score ≥ 10.

**Table 4 ijerph-17-08615-t004:** Determinants of burnout, anxiety and depression, univariate linear models. Only significant coefficients and variable levels are reported.

	Burnout		Anxiety		Depression	
	B (SE)	t	B (SE)	t	B (SE)	t
**Gender** (ref. male)						
Female	3.78 (1.77)	2.13 *	1.65 (5.22)	3.17 **	1.48 (.43)	3.42 **
**Professional role** (ref. psychologist)						
Medical doctor	10.25 (3.68)	2.78 **				
**Level of contact** (ref. no direct contacts)						
Had contacts with non-infected users	4.42 (2.12)	2.08 *				
Had contacts with infected users					−1-11 (.56)	−2.00 *
**Setting** (reference inpatient)						
Outpatient service	4.83 (2.11)	2.29 *				
**Perceived risk of contracting covid-19 at work** (ref. low probability)						
Medium probability	5.83 (1.76)	3.22 **	1.30 (.52)	2.51 *		
High probability	7.00 (2.71)	2.57 *	2.02 (.81)	2.47 *	1.35 (.67)	1.99 *

* *p* = < 0.05; ** = *p* < 0.01; The following variables were included in the models but are not displayed because no significant effects were found: age (covariate), availability of PPE, perceived change in workload, availability of staff psychological support.

**Table 5 ijerph-17-08615-t005:** Interventions performed by mental health workers in outpatient and inpatient services during the early stage of Covid-19 outbreak.

	Suspended Due to the Emergency		Direct Contacts Only		Distance Contacts Only		Both Direct and Distance		Not my Task or Duty	
**Individual counselling sessions ****										
Outpatient	2	1.3%	4	2.6%	90	58.4%	49	31.8%	9	5.8%
Inpatient	1	1.1%	48	55.2%	5	5.7%	19	21.8%	13	14.9%
**Psychotherapy ****										
Outpatient	5	3,2%	1	0.6%	52	33.8%	12	7.8%	83	53.9%
Inpatient	3	3.4%	5	5.7%	2	2.3%	2	2.3%	74	85.1%
**Meetings with family members ***										
Outpatient	15	9.7%	5	3.2%	82	53.2%	24	15.6%	28	18.2%
Inpatient	8	9.2%	7	8.0%	36	41.4%	7	8.0%	28	32.2%
**Individual psychosocial rehabilitation ****										
Outpatient	5	3.2%	6	3.9%	42	27.3%	11	7.1%	89	57.8%
Inpatient	2	2.3%	34	39.1%	4	4.6%	8	9.2%	38	43.7%
**Group psychosocial rehabilitation ****										
Outpatient	43	27.9%	1	0.6%	33	21.4%	3	1.9%	74	48.1%
Inpatient	9	10.3%	39	44.8%	4	4.6%	5	5.7%	29	33.3%
**Drug administration ****										
Outpatient	2	1,3%	12	7.8%	7	4.5%	6	3.9%	126	81.8%
Inpatient	0	0,0%	33	37.9%	2	2.3%	2	2.3%	49	56.3%
**Assistance with personal hygiene and meals ****										
Outpatient	6	3.9%	4	2.6%	5	3.2%	1	0.6%	137	89.0%
Inpatient	1	1.1%	43	49.4%	1	1.1%	1	1.1%	40	46.0%
**Visits in the community ****										
Outpatient	39	25.3%	9	5.8%	0	0.0%	1	0.6%	104	67.5%
Inpatient	35	40.2%	14	16.1%	0	0.0%	2	2.3%	35	40.2%
**Home visits**										
Outpatient	40	26.0%	18	11.7%	5	3.2%	3	1.9%	87	56.5%
Inpatient	30	34.5%	4	4.6%	1	1.1%	0	0.0%	51	58.6%
**Back office or supervision ****										
Outpatient	9	5.8%	9	5.8%	49	31.8%	9	5.8%	77	50.0%
Inpatient	7	8.0%	17	19.5%	3	3.4%	10	11.5%	49	56.3%

* *p* = < 0.05; ** = *p* < 0.01.

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
