# Peer review of "The Early Impact of the Covid-19 Emergency on Mental Health Workers: A Survey in Lombardy, Italy"

_ijerph, 2020, doi:10.3390/ijerph17228615_

Round 1
Reviewer 1 Report
Interesting paper, the differences between in distress levels in staff working patient and outpatient settings is interesting although the potential explanations in terms of direct patient contact and risk exposure to some extent probably explain this. Nonetheless it is helpful to have this demonstrated.
The perception of increased work load in the inpatient setting is also interesting which contrasts with the reduction in inpatient admissions noted in other papers. Perhaps a higher level of acuity in the admitted patients during the lockdown may explain the disparity?
A comment on the methodology: How representative of mental health staff working in Lombardy is the sample you obtained? The information presented on this at the moment is insufficient.
Also, this was not a random sample; could you add some information on the validity of the method you chose in terms of managing selection biases?
Several language and spelling error need correction.
22 professionals
25 participants
42 approximately
Line 130 and 131 need to be amended as it does not make sense. Suggest: The inclusion of subjects was conditional on them having read the Information....
144 "participants"
161 "or that they did not" and "declared that they were inadequately"
199 a high
213 despite the emergency
222 in person
231/232 during the outbreak in inpatient setting diverges from what Clerici and colleagues...
255/256 medical doctors seem
290 should be assessed
Author Response
Interesting paper, the differences between in distress levels in staff working patient and outpatient settings is interesting although the potential explanations in terms of direct patient contact and risk exposure to some extent probably explain this. Nonetheless it is helpful to have this demonstrated.
We thank the reviewer for the acknowledgement.
The perception of increased work load in the inpatient setting is also interesting which contrasts with the reduction in inpatient admissions noted in other papers. Perhaps a higher level of acuity in the admitted patients during the lockdown may explain the disparity?
The higher level of patients’ acuity may be an additional explanation that we didn’t take into account. We could add it in the discussion section.
A comment on the methodology: How representative of mental health staff working in Lombardy is the sample you obtained? The information presented on this at the moment is insufficient.
Our sample presents all the professional categories of mental health workers in Lombardy, including staff working in private and non profit organizations that manage a relevant part of mental health services. In a previous survey from 2015 (Rapisarda et al., 2019, Development and validation etc…, added in the references), based on a similar sample of mental health workers, the first Author found a similar distribution of professional categories with significantly fewer psychologists and peer support workers. However, while the amount of peer support workers in our sample is overrepresented and could be considered as consequence of a sampling bias due to the involvement of a peer support association, the relative larger number of psychologists compared to the 2015 is a consequence of the choice to include in the 2020 survey also freelance professional that work in private practice.
We included some more specification about sampling in discussion section and added the Rapisarda et al, 2019 reference.
Also, this was not a random sample; could you add some information on the validity of the method you chose in terms of managing selection biases?
To increase the validity of the sampling method, we adopted two strategies. We started by sending the survey invitation to directors and service coordinators that could spread the survey invitation to multiprofessional groups from public, non profit or private services. Then, after the first week of data collection, we compared our sample with the 2015 survey and we tried to correct and compensate over and under represented professional categories by sending targeted invitation to specific professional groups (i.e. nurses). Although we couldn’t eliminate completely some forms of sampling bias, we obtained an heterogeneous sample that included significant members from every professional groups.
Several language and spelling error need correction.
22 professionals
25 participants
42 approximately
Line 130 and 131 need to be amended as it does not make sense. Suggest: The inclusion of subjects was conditional on them having read the Information....
144 "participants"
161 "or that they did not" and "declared that they were inadequately"
199 a high
213 despite the emergency
222 in person
231/232 during the outbreak in inpatient setting diverges from what Clerici and colleagues...
255/256 medical doctors seem
290 should be assessed
We wish to thank the reviewer for the proofreading of the manuscript. We corrected all the typos and grammar mistakes reported above.
Reviewer 2 Report
Article of interest to readers and well written. I would only ask for small changes and clarifications in the methodology section and tables.
In Section 2.1 Design and procedure: it is indicated that a snowball dissemination strategy was carried out. But it could be interesting to know the approximate number of the study population in each subgroup. In order to know if any of them are underrepressented. We know in the results section the proportion of each group in the analysed sample but we do not know what the population is.
We don´t know if socio-demographic and professional background section and work conditions during the emergency questions have been obtained fro a previous publication (source not indicated) or if they have had a pilot study to verify it.
Tables should be checked because some signs are out of place.
Author Response
Article of interest to readers and well written. I would only ask for small changes and clarifications in the methodology section and tables.
We thank the reviewer for the acknowledgement.
In Section 2.1 Design and procedure: it is indicated that a snowball dissemination strategy was carried out. But it could be interesting to know the approximate number of the study population in each subgroup. In order to know if any of them are underrepressented. We know in the results section the proportion of each group in the analysed sample but we do not know what the population is.
Our sample presents all the professional categories of mental health workers in Lombardy, including staff working in private and non profit organizations that manage a relevant part of mental health services. In a previous survey from 2015 (Rapisarda et al., 2019, Development and validation etc…, added in the references), based on a similar sample of mental health workers, the first Author found a similar distribution of professional categories with significantly fewer psychologists and peer support workers. However, while the amount of peer support workers in our sample is overrepresented and could be considered as consequence of a sampling bias due to the involvement of a peer support association, the relative larger number of psychologists compared to the 2015 is a consequence of the choice to include in the 2020 survey also freelance professional that work in private practice.
We included some more specification about sampling in discussion section and added the Rapisarda et al, 2019 reference.
We don´t know if socio-demographic and professional background section and work conditions during the emergency questions have been obtained fro a previous publication (source not indicated) or if they have had a pilot study to verify it.
Socio-demographic and professional background questions were reprised from a previous survey made in 2015 (Rapisarda et al., 2019, Development and validation etc…, added in the references). Work conditions during the emergency questions have been created ad hoc for the present study, so no previous validation was performed. We added some additional specification in the method section.
Tables should be checked because some signs are out of place.
We attempted to check tables, now all the signs and cells should be in place.
Reviewer 3 Report
The authors investigated the impact of the covid-19 pandemic and quarantine on the well-being and work practices of mental health professionals. This is a study that would be of interest to a wide range of professionals and lay persons. It is very well written and well organized. The statistical methods are appropriate and the results are clear. The authors end with a very focused and sufficiently detailed Discussion. There are a few minor grammatical changes that would need to be made but other than that, the quality of the English is great!
p. 3, line 134: Change to '40 (11.9%) declared that they suspended'..
p. 3, line 137: Change to 'dropped out'...
p. 4, line 161: Change to 'workers (21.5%) declared that they were inadequately equipped.'
p. 9, line 222: Change to 'delivered mostly in person'.
p. 10, line 256: Change to 'medical doctors seemed to be...'
Author Response
The authors investigated the impact of the covid-19 pandemic and quarantine on the well-being and work practices of mental health professionals. This is a study that would be of interest to a wide range of professionals and lay persons. It is very well written and well organized. The statistical methods are appropriate and the results are clear. The authors end with a very focused and sufficiently detailed Discussion. There are a few minor grammatical changes that would need to be made but other than that, the quality of the English is great!
We sincerely thank the reviewer for the very positive feedback, we appreciate your interest in our manuscript!
p. 3, line 134: Change to '40 (11.9%) declared that they suspended'..
p. 3, line 137: Change to 'dropped out'...
p. 4, line 161: Change to 'workers (21.5%) declared that they were inadequately equipped.'
p. 9, line 222: Change to 'delivered mostly in person'.
p. 10, line 256: Change to 'medical doctors seemed to be...'
We wish to thank the reviewer for the proofreading of the manuscript. We corrected all the typos and grammar mistakes reported above.